# A single-cell atlas of the mouse and human prostate reveals heterogeneity and conservation of epithelial progenitors

Laura Crowley[1,2,3,4,5†], Francesco Cambuli[1,2,3,4,5†], Luis Aparicio[4,5,6†], Maho Shibata[1,2,3,4,5‡], Brian D Robinson[7], Shouhong Xuan[1,2,3,4,5], Weiping Li[1,2,3,4,5], Hanina Hibshoosh[5,8], Massimo Loda[7], Raul Rabadan[4,5,6*], Michael M Shen[1,2,3,4,5*]

[1]Department of Medicine, Columbia University Irving Medical Center, New York, United States; [2]Department of Genetics and Development, Columbia University Irving Medical Center, New York, United States; [3]Department of Urology, Columbia University Irving Medical Center, New York, United States; [4]Department of Systems Biology, Columbia University Irving Medical Center, New York, United States; [5]Herbert Irving Comprehensive Cancer Center, Columbia University Irving Medical Center, New York, United States; [6]Department of Biomedical Informatics, Columbia University Irving Medical Center, New York, United States; [7]Department of Pathology and Laboratory Medicine, Weill Medical College of Cornell University, New York, United States; [8]Department of Pathology and Cell Biology, Columbia University Irving Medical Center, New York, United States

*For correspondence:
rr2579@cumc.columbia.edu (RR);
mshen@columbia.edu (MMS)

†These authors contributed equally to this work

Present address: ‡Department of Anatomy and Cell Biology, School of Medicine and Health Sciences, The George Washington University Cancer Center, The George Washington University, Washington, DC, United States

Competing interests: The authors declare that no competing interests exist.

**Abstract** Understanding the cellular constituents of the prostate is essential for identifying the cell of origin for prostate adenocarcinoma. Here, we describe a comprehensive single-cell atlas of the adult mouse prostate epithelium, which displays extensive heterogeneity. We observe distal lobe-specific luminal epithelial populations (LumA, LumD, LumL, and LumV), a proximally enriched luminal population (LumP) that is not lobe-specific, and a periurethral population (PrU) that shares both basal and luminal features. Functional analyses suggest that LumP and PrU cells have multipotent progenitor activity in organoid formation and tissue reconstitution assays. Furthermore, we show that mouse distal and proximal luminal cells are most similar to human acinar and ductal populations, that a PrU-like population is conserved between species, and that the mouse lateral prostate is most similar to the human peripheral zone. Our findings elucidate new prostate epithelial progenitors, and help resolve long-standing questions about anatomical relationships between the mouse and human prostate.

## Introduction

Significant anatomical differences between the mouse and human prostate have long hindered analyses of mouse models of prostate diseases. The mouse prostate can be separated into anterior (AP), dorsal (DP), lateral (LP), and ventral (VP) lobes; the mouse dorsal and lateral lobes are often combined as the dorsolateral prostate (DLP) (*Cunha et al., 1987*; *Shappell et al., 2004*; *Shen and Abate-Shen, 2010*). In contrast, the human prostate lacks defined lobes, and instead is divided into different histological zones (central, transition, and peripheral); the peripheral zone represents the predominant site of prostate adenocarcinoma, whereas benign prostatic hyperplasia (BPH) occurs in the transition zone (*Cunha et al., 2018*; *Ittmann, 2018*; *Shappell et al., 2004*). Moreover, unlike the mouse, the human prostate has distinct ductal and acinar regions. Although microarray gene expression profiling has suggested that the DLP is most similar to the human peripheral zone

(*Berquin et al., 2005*), there is no consensus on the relationship between mouse lobes and human zones (*Ittmann, 2018*; *Ittmann et al., 2013*; *Shappell et al., 2004*).

The adult prostate epithelium is comprised of luminal, basal, and rare neuroendocrine cells (*Shen and Abate-Shen, 2010*; *Toivanen and Shen, 2017*), and cellular heterogeneity has been suggested within the luminal (*Barros-Silva et al., 2018*; *Chua et al., 2014*; *Karthaus et al., 2020*; *Karthaus et al., 2014*; *Kwon et al., 2016*; *Liu et al., 2016*) and basal compartments (*Goldstein et al., 2008*; *Lawson et al., 2007*; *Wang et al., 2020*). Lineage-tracing analyses have shown that the hormonally intact adult prostate epithelium is maintained by unipotent progenitors within the basal and luminal epithelial compartments (*Choi et al., 2012*; *Lu et al., 2013*; *Wang et al., 2013*). However, following tissue dissociation, both basal and luminal cells can act as bipotent progenitors in organoid or tissue reconstitution assays (*Chua et al., 2014*; *Karthaus et al., 2014*). The progenitor properties of basal cells may reflect their ability to generate luminal progeny during tissue repair after wounding or inflammation (*Kwon et al., 2014*; *Toivanen et al., 2016*), but the role of presumptive luminal progenitors has been less clear. In particular, several studies have suggested increased progenitor potential in the proximal region of the prostate (nearest to the urethra) (*Burger et al., 2005*; *Goto et al., 2006*; *Tsujimura et al., 2002*), particularly for proximal luminal cells (*Guo et al., 2020*; *Karthaus et al., 2020*; *Kwon et al., 2016*; *Wei et al., 2019*; *Zhang et al., 2018*). However, the nature and distribution of these epithelial populations have been poorly characterized.

## Results

### Distinct luminal epithelial populations in the mouse prostate

To examine cellular heterogeneity, we performed single-cell RNA-sequencing of whole prostates from adult wild-type mice at 10 weeks of age. We microdissected the full proximal-distal extent of each prostate lobe down to its junction with the urethral epithelium (*Figure 1—figure supplement 1A–C*). We noted that the anterior (AP), dorsal (DP), and lateral (LP) lobes joined the urethra in close proximity on the dorsal side, whereas the ventral lobe (VP) had a distinct junction ventrally. As previously described (*Cunha et al., 1987*; *Shappell et al., 2004*; *Shen and Abate-Shen, 2010*), each lobe has a characteristic morphology, pattern of ductal branching, and histological appearance (*Figure 1—figure supplement 1D–F*). Single-cell populations were obtained using a multi-step tissue dissociation protocol with successive enzymatic digestion, trituration, and filtering steps. Droplet-based single-cell RNA-sequencing was performed on the 10x Genomics Chromium platform (see Materials and methods for details).

For analysis of the resulting data, we performed batch effect correction if needed for aggregation of datasets (*Stuart et al., 2019*), followed by application of *Randomly*, an algorithm that uses random matrix theory to reduce noise in single-cell datasets (*Aparicio et al., 2020*). Using the universality property of random matrix theory on eigenvalues and eigenvectors of sparse matrices, *Randomly* discriminates biological signals from noise and sparsity-induced confounding signals, which typically comprise approximately 98% of the data, based on a survey of published single-cell datasets (*Aparicio et al., 2020*). The *Randomly* algorithm is based on the three-fold structure of a single-cell dataset: a random matrix (95% or more), a sparsity-induced (fake) signal, and a biological signal. The algorithm uses the universality properties of random matrix theory for both eigenvalues and eigenvectors to detect the biological signal. After de-noising of single-cell data, we performed clustering using the Leiden algorithm as implemented in *Wolf et al., 2018*, with selection of the number of clusters based on the mean silhouette score. Processing by *Randomly* followed by dimensional reduction for visualization using t-SNE (t-distributed Stochastic Neighbor Embedding) or UMAP (Uniform Manifold Approximation and Projection) plots facilitated the identification of cell populations with distinct transcriptional signatures (*Figure 1—figure supplement 2*). Additional description of computational methods is provided in Materials and methods.

We identified distinct luminal, basal, and neuroendocrine populations that were annotated based on the expression of marker genes, as visualized in an aggregated dataset composed of 5288 cells from two whole prostates (tSNE plot shown in *Figure 1A,D*; UMAP plot shown in *Figure 1—figure supplement 3A*). Notably, we could identify five different luminal epithelial populations, a single basal population, rare neuroendocrine cells, and a small population of epithelial cells that expresses

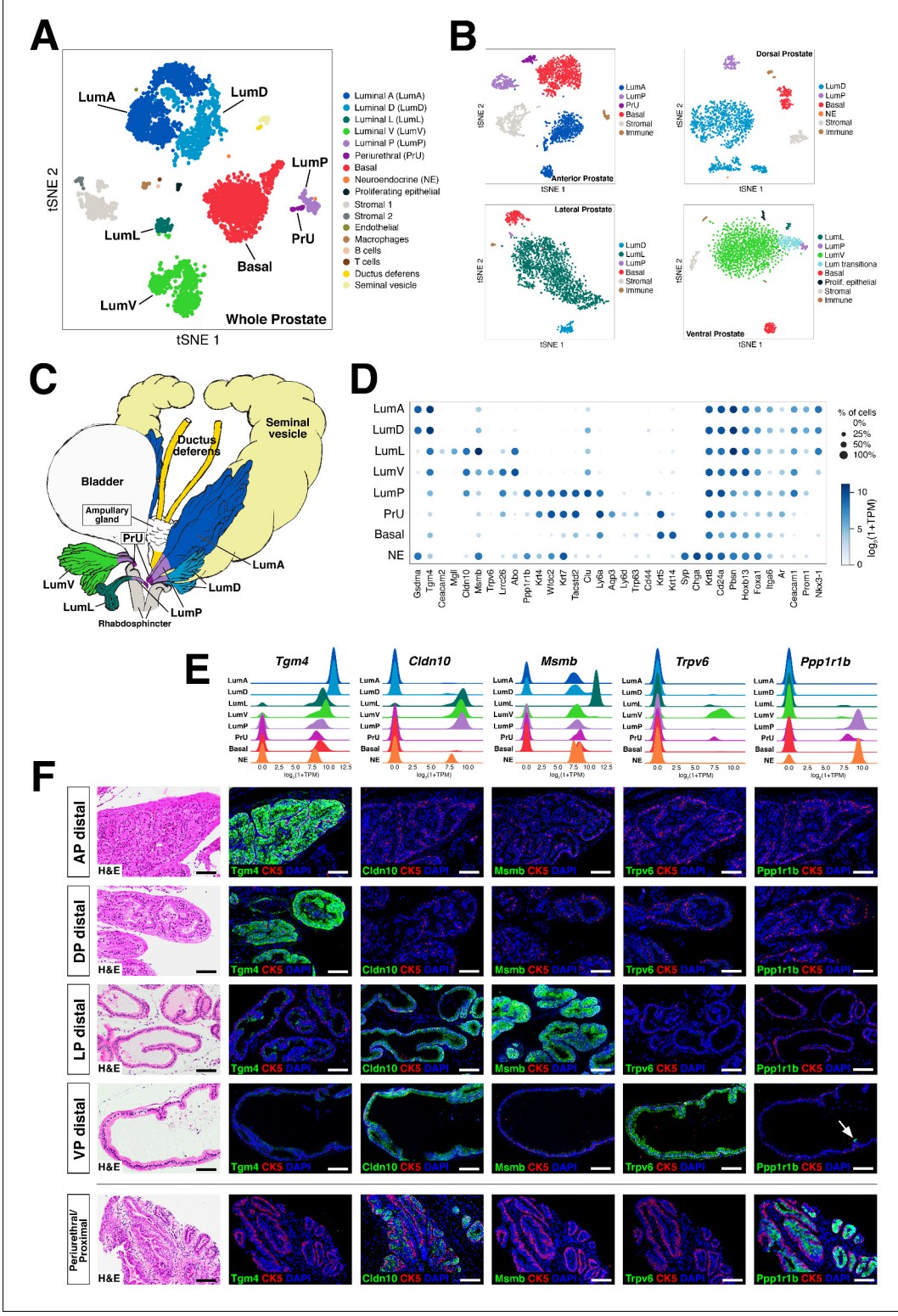

**Figure 1.** Single-cell analysis identifies prostate luminal epithelial heterogeneity. (**A**) *t*-distributed stochastic neighbor embedding (tSNE) plot of 5288 cells from an aggregated dataset of two normal mouse prostates, processed by *Randomly* and clustered using the Leiden algorithm. (**B**) tSNE representation of each prostate lobe (AP: 2735 cells; DP: 1781 cells; LP: 2044 cells; VP: 1581 cells). (**C**) Schematic model of prostate lobes with the urethral rhabdosphincter partially removed, with the distribution of luminal epithelial populations indicated. (**D**) Dot plot of gene expression levels in each epithelial population for selected marker genes. (**E**) Ridge plots of

*Figure 1 continued on next page*

*Figure 1 continued*

marker genes showing expression in each population. (F) Hematoxylin-eosin (H and E) and immunofluorescence (IF) images of selected markers in serial sections; the periurethral/proximal region shown is from the AP and DP. Arrow in VP distal indicates distal cell with *Ppp1r1b* expression. Scale bars indicate 50 µm.

The online version of this article includes the following figure supplement(s) for figure 1:

**Figure supplement 1.** Anatomy and dissection of mouse prostate lobes.
**Figure supplement 2.** Random-matrix analysis of single-cell datasets.
**Figure supplement 3.** UMAP plots of mouse single-cell RNA-seq data.
**Figure supplement 4.** Dot plot of expression levels for selected genes in each epithelial population.
**Figure supplement 5.** Additional marker validation for epithelial populations.

both basal and luminal markers. We could also identify distinct stromal and immune components, corresponding to two different stromal subsets (*Kwon et al., 2019*), as well as immune cells (macrophages, T cells, B cells); some datasets also contained small populations of contaminating vas deferens and seminal vesicle cells.

To assess whether some of these epithelial populations might be lobe-specific, we next performed single-cell RNA-seq analyses of individual lobes (tSNE plots shown in *Figure 1B*; UMAP plots shown in *Figure 1—figure supplement 3B*). We found that four of the luminal populations identified in the aggregated dataset were highly lobe-specific; hence, we named these populations LumA (AP-specific), LumD (DP-specific), LumL (LP-specific), and LumV (VP-specific). The remaining luminal population was observed in the datasets for all four lobes and was highly enriched in the proximal portion of each lobe; thus, we termed this population LumP (proximal) (*Figure 1C*).

## Spatial localization and morphology of epithelial populations

To examine the lobe-specificity and spatial distribution of these luminal populations, we identified candidate markers based on gene expression patterns in our single-cell datasets (marker genes of interest and/or used in this study are shown in the dot plot of *Figure 1D*; a larger set of differentially expressed genes are shown in *Figure 1—figure supplement 4*). We focused our analyses on candidate markers with commercially available antibodies that displayed low background staining in control sections, and selected antibodies for further study that were specific for the cell populations predicted by single-cell analyses (*Figure 1D–F*; *Figure 1—figure supplement 5*). For example, we found that the LumA and LumD marker *Tgm4* was highly expressed by luminal cells in the distal region of the AP and DP, and that the LumL marker *Msmb* marked distal luminal cells in the LP. In contrast, the LumP marker *Ppp1r1b* was highly enriched in luminal cells that were primarily found in the proximal regions of all four lobes (*Figure 1D–F*). However, more general luminal markers such as androgen receptor (*Ar*) were expressed in all luminal populations (*Figure 1—figure supplement 5*).

Next, we investigated the spatial localization of these luminal populations along the proximal-distal axis in each lobe. We found that the LumP-containing proximal region extended from inside the rhabdosphincter to the first major ductal branch point in the AP, DP, and VP, but not the LP, whereas the bulk of the lobes corresponded to distal regions (*Figure 1C*). In the AP and DP, we found a discrete boundary in the medial region between the proximal LumP population and distal LumA or LumD populations, respectively (*Figure 2A,C*; *Figure 2—figure supplement 1A*). In contrast, the LP had a population between the proximal and distal regions that expressed low levels of LumL markers (*Figure 2A*; *Figure 2—figure supplement 1A*). Histological analyses revealed that distal luminal cells of each lobe had a tall columnar appearance consistent with secretory function, whereas proximal LumP cells typically had a cuboidal morphology. Notably, this analysis also revealed heterogeneity in the distal region of each lobe, with rare clusters of 1–10 LumP cells observed in the distal AP, DP, and LP, but more frequent LumP clusters in the distal VP (*Figure 2B*).

To clarify the phenotypic differences between proximal and distal luminal populations, we performed scanning electron microscopy of an 8-week-old anterior lobe (*Figure 2C*; *Figure 2—figure supplement 1B*). LumA cells displayed dense regions of rough endoplasmic reticulum throughout the cytoplasm, many free ribosomes, and abundant secretory vesicles on the apical surface, typical of secretory cells. In contrast, LumP cells displayed areas of high mitochondrial density, complex membrane interdigitation, and no vesicles. At the proximal-distal boundary, we observed an abrupt transition between cellular morphologies that took place within 1–2 cell diameters. These

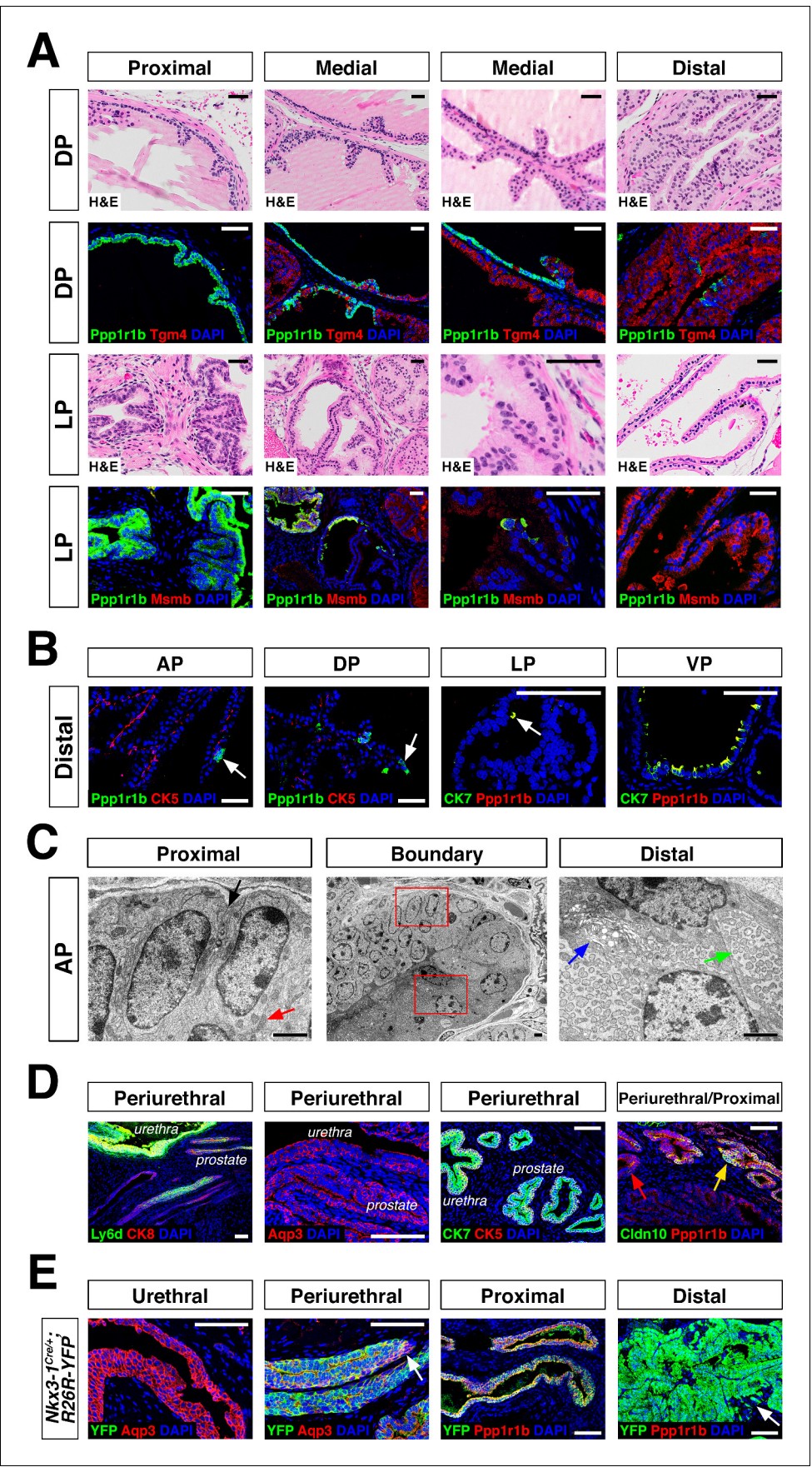

**Figure 2.** Luminal epithelial populations display spatial and morphological heterogeneity. (**A**) H and E and IF of serial sections from the DP and LP, showing expression of proximal (*Ppp1r1b*) and distal (*Tgm4, Msmb*) markers; note apparent differences in the boundary regions of the two lobes. (**B**) Detection of distally localized LumP cells (arrows) in all four lobes; these are most abundant in the VP. (**C**) Scanning electron micrographs of the boundary region of the AP; central low-power image is flanked by high-power images of boxed regions. Red arrow, mitochondria; black arrow, membrane interdigitation; blue arrow, Golgi apparatus; green arrow, rough endoplasmic reticulum. (**D**) Identification of the periurethral region. Cells in the periurethral region generally express Ly6d, Ck7, Aqp3, and Ppp1r1b; notably, Cldn10-expressing LumP cells decrease approaching the periurethral region (**E**) Lineage-marking in *Nkx3.1^{Cre/+}; R26R-YFP* mice (n = 3) shows widespread YFP expression in the periurethral, proximal, and distal AP; small patches remain unrecombined and lack YFP (arrows). Scale bars in (**A,B,D,E**) indicate 50 µm; scale bars in (**C**) indicate 2 µm.

The online version of this article includes the following figure supplement(s) for figure 2:

**Figure supplement 1.** Additional analysis of proximal-distal heterogeneity.

---

ultrastructural differences indicate that the LumA and LumP populations represent distinct cell types, rather than cell states.

Next, we investigated the remaining epithelial population, which shares basal and luminal features in our single-cell RNA-seq analysis (*Figure 1A*). We found that this small population was enriched in the most proximal region of all four lobes, residing inside the rhabdosphincter and adjacent to the urethral junction (*Figure 2D*; *Figure 1—figure supplement 1C*); hence, we termed this novel population PrU (periurethral). Although this PrU population co-expresses some markers with LumP (*Figure 1D*), it also expresses several urothelial markers such as *Ly6d* and *Aqp3* (*Figure 2D*).

The proximity of the PrU and LumP populations to the urethra and their co-expression of multiple markers led us to investigate their developmental origin. Consequently, we examined the expression of *Nkx3-1*, whose mRNA expression marks epithelial cells in ductal derivatives of the developing urogenital sinus, such as the prostate, but not the urothelium (*Bhatia-Gaur et al., 1999*); similarly, the *Nkx3-1^{Cre}* driver also marks early prostate bud cells but not the urogenital sinus during development (*Thomsen et al., 2008*; *Zhang et al., 2008*). In the adult prostate, *Nkx3-1* is expressed by all four distal luminal populations (LumA, LumD, LumL, LumV), but is not expressed by LumP or PrU (*Figure 1—figure supplement 5*). However, we found the *Nkx3-1^{Cre}* driver lineage-marks most of the cells in all of these populations, including LumP and PrU, but not the urethra (*Figure 2E*). These data indicate that the LumP and PrU populations are derived from *Nkx3-1* expressing prostate epithelial cells and are distinct from the urothelium.

## Functional analysis of epithelial populations

We used an approach based on optimal transport theory to ascertain the relationships of these prostate epithelial populations (see Materials and methods). We calculated the similarity among cell populations in mouse using the Wasserstein-1 distance as a measure for phenotypic distance among cell populations, based on clusters in the latent space previously obtained by the *Randomly* algorithm. The pair-wise comparisons between populations (*Figure 3A*) can be captured by a neighbor-joining tree (*Figure 3B*), in which lower Wasserstein distance indicates greater similarity. We found that the distal populations grouped together, with the LumA and LumD populations being most closely related, followed by LumL and LumV. These distal populations were next most closely related to LumP, which in turn was most similar to PrU, followed by basal cells, suggesting a lineage relationship between LumP and distal luminal populations.

To investigate the functional properties of each epithelial population, we developed a flow sorting strategy to isolate these populations (*Figure 3C*; *Figure 3—figure supplement 1A*); we performed re-sorting experiments to assess the purity of the isolated populations (*Figure 3—figure supplement 1B*). We performed organoid formation assays with isolated cell populations using the defined ENR-based medium (*Drost et al., 2016*; *Karthaus et al., 2014*) as well as hepatocyte medium (HM), which has a more complex composition including serum (*Chua et al., 2014*). Despite differences in overall efficiency between media conditions, we consistently found that the PrU and basal populations were most efficient at forming organoids, followed by LumP, whereas the efficiency of distal LumA, LumD, LumL, and LumV was significantly lower (*Figure 3D,E*).

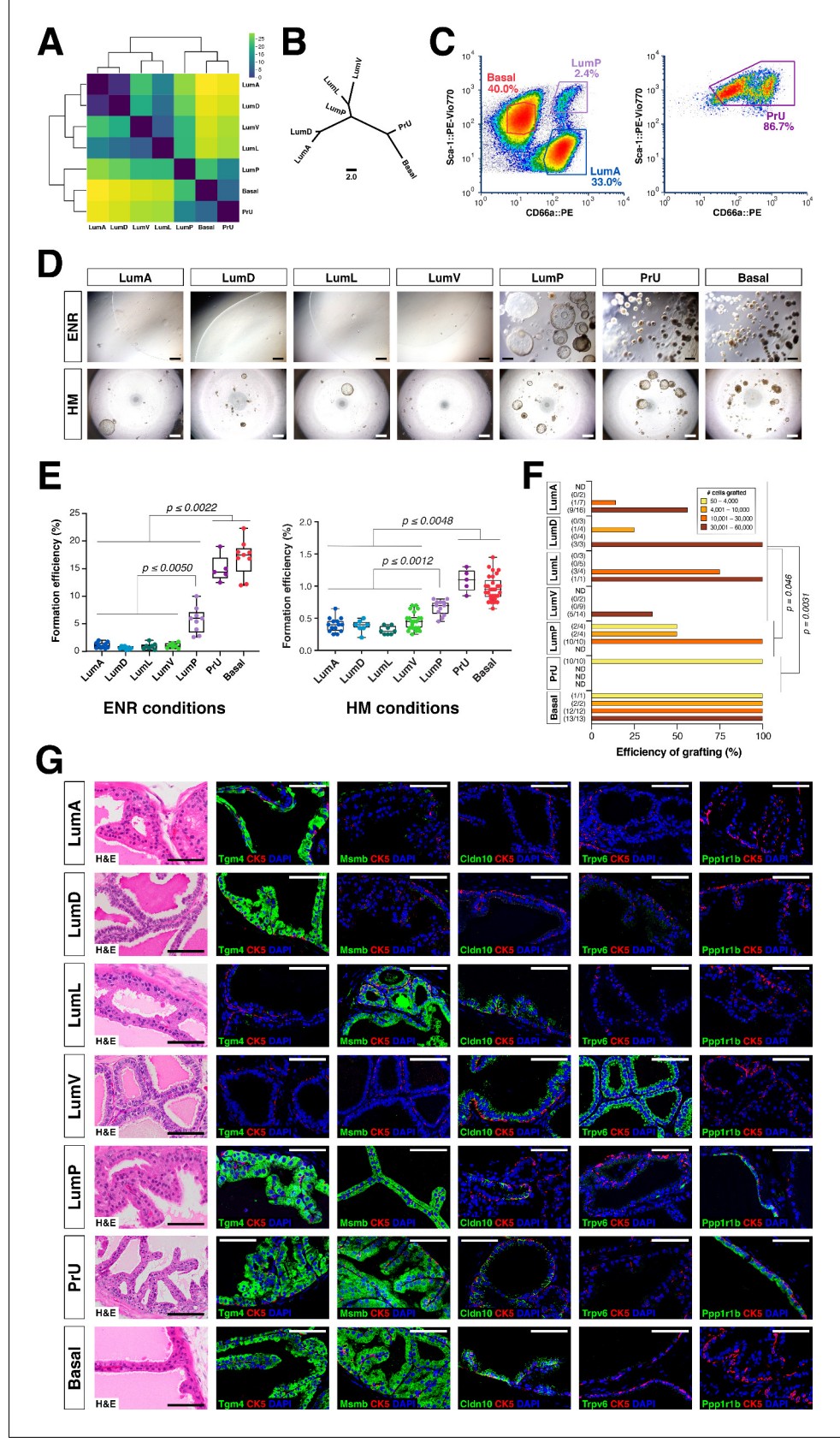

**Figure 3.** Functional analysis of epithelial populations in organoid and tissue reconstitution assays. (**A**) Heatmap visualization of the Wasserstein distances between epithelial populations with hierarchical clustering. (**B**) Tree visualization of Wasserstein distances. (**C**) Flow sorting of distinct epithelial populations from the AP lobe. (**D**) Organoids grown from sorted epithelial cells in two distinct culture conditions. ENR conditions: sorted cells from *UBC-GFP* mice plated at 1000 cells/well, imaged at day 10. Hepatocyte Media (HM) conditions: sorted cells from wild type C57BL/6 mice plated at 2000-5000 cells/well and imaged on day 12-13. (**E**) Organoid formation efficiency plots. Maximum p-values for each pair-wise comparison are indicated. (**F**) Grafting efficiency in tissue reconstitution assays (average p-value shown). LumP, PrU, and basal are significantly more efficient at generating grafts from smaller number of cells relative to distal luminal populations. (**G**) H&E and IF of sections from fully-differentiated renal grafts; positive staining corresponds to results found in $\geq 3$ independent grafts. Scale bars indicate 50 µm.

The online version of this article includes the following figure supplement(s) for figure 3:

**Figure supplement 1.** Flow-sorting strategy and validation.
**Figure supplement 2.** Additional marker analysis of renal grafts.

Next, we assessed the progenitor potential of isolated epithelial populations using in vivo tissue reconstitution assays. These assays involve recombination of dissociated epithelial cells with rat embryonic urogenital mesenchyme followed by renal grafting, and have been extensively utilized for analysis of progenitor properties in the prostate (*Lawson et al., 2007*; *Wang et al., 2013*; *Xin et al., 2003*). We observed significant variation in the frequency of graft formation depending on the number and type of input epithelial cells (*Figure 3F*). Based on histological and immunostaining analyses, we found that each epithelial population typically gave rise to cells of the same type, but their ability to generate cells of other populations varied considerably (*Figure 3G*; *Figure 3—figure supplement 2A*). Notably, grafts using distal luminal cells required relatively large numbers of input cells to grow (approximately 30,000 cells) (*Figure 3F*), and were mostly composed of the same population (i.e. LumA cells generated LumA cells) together with a normal or reduced percentage of basal cells; additionally, these grafts could contain small patches of cells expressing LumP markers on their periphery (*Figure 3G*; *Figure 3—figure supplement 2A*). Interestingly, LumV cells had the lowest grafting efficiency and generally formed small ductal structures that lacked basal cells. In contrast, LumP, PrU, and basal cells could produce grafts with significantly lower input cell numbers (approximately 1,000 cells), which contained LumP cells together with multiple distal luminal populations as well as a normal ratio of basal cells. We excluded contribution of host cells and rat urogenital epithelium to grafts using control tissue reconstitution assays with GFP-expressing donor epithelial cells (*Figure 3—figure supplement 2B*).

Taken together, these results suggest a spectrum of progenitor potential among the different epithelial populations. Both PrU and basal cells possessed high progenitor activity in these assays, with LumP cells also displaying enhanced activity. In contrast, the distal luminal populations are much less efficient in these assays. These findings are consistent with the inferred relationships between these populations based on molecular (*Figure 3A,B*) as well as histological and ultrastructural analyses (*Figure 2A,C*).

## Comparison of human and mouse prostate epithelial populations

To examine the conservation of epithelial populations between the mouse and human prostate, we performed single-cell RNA-seq analyses of tissue samples from human prostatectomies (*Figure 4—source data 1*). We identified two distinct luminal populations as well as a PrU-like population (tSNE plots shown in *Figure 4A–C*; UMAP plots shown in *Figure 4—figure supplement 1*), and examined their spatial distribution by immunostaining of benign human prostate tissue (*Figure 4—source data 1*). Using specific markers, we found that one luminal population (Lum Ductal, expressing KRT7 and RARRES1) was primarily localized to ducts, whereas the second population was predominantly acinar (Lum Acinar, expressing MSMB and MME), although some intermixing could be observed within ducts (*Figure 4G*). As in the mouse, the PrU-like population expressed both basal and luminal genes; similarly, some PrU-like markers were shared with Lum Ductal.

To extend this analysis, we computed the Wasserstein distances for each cross-species pair-wise comparison between epithelial and major non-epithelial populations (*Figure 4D–F*). This analysis showed that a PrU-like population is conserved between species. Furthermore, the human Lum

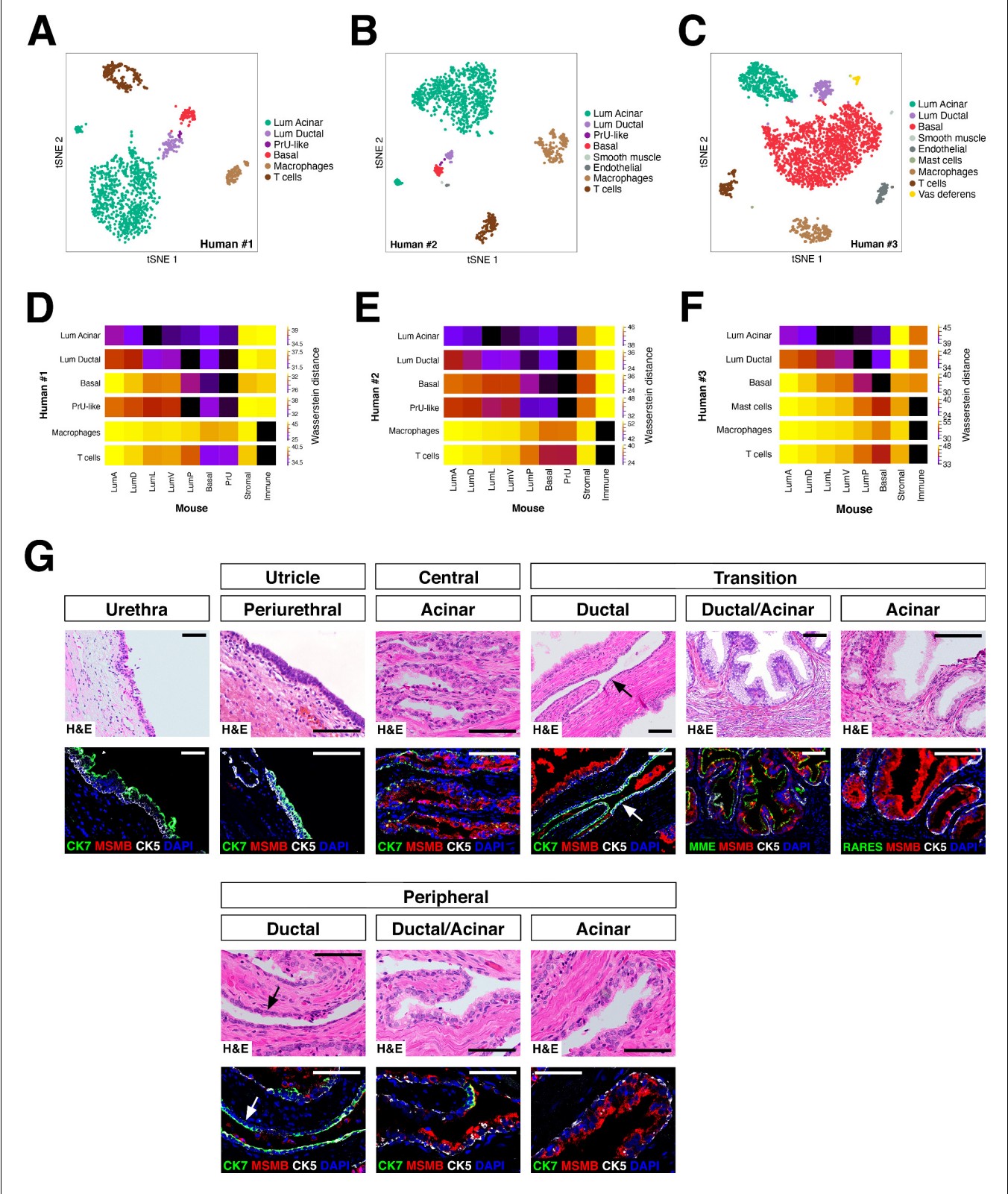

**Figure 4.** Heterogeneity and conservation of luminal populations in the human prostate. (A–C) tSNE plot of scRNA-seq data (A, 1600 cells; B, 2303 cells; C, 2825 cells) from three independent human prostatectomy samples. (D–F) Heatmap visualization of Wasserstein distances between the human and mouse prostate populations for each dataset. (G) H and E and IF images of serial sections from human prostate, showing regions of the prostatic utricle, central, transition, and peripheral zones. Arrows indicate regions of ductal morphology. Scale bars indicate 50 μm.

*Figure 4 continued on next page*

*Figure 4 continued*

The online version of this article includes the following source data and figure supplement(s) for figure 4:

**Source data 1.** Human prostate samples and corresponding clinical data.

**Figure supplement 1.** UMAP plots of human single-cell RNA-seq data.

Ductal cells are most closely related to mouse LumP, whereas the Lum Acinar cells are most closely related to LumL followed by LumV. Notably, these relationships were observed in each dataset.

## Discussion

We have generated a comprehensive cellular atlas of the prostate epithelium and have defined spatial, morphological, and functional properties of each epithelial population. Our analyses have revealed spatial and functional heterogeneity primarily in the luminal epithelial compartment, including distinct cell populations along the proximal-distal axis as well as lobe-specific identities. Notably, we have shown marked differences in progenitor potential between cell identities, which likely correspond to distinct cell types rather than cell states (*Morris, 2019*). In tissue reconstitution assays, the ability of LumP and PrU cells to generate luminal distal and basal cells suggests that both populations have properties of multipotent progenitors. In contrast to the luminal compartment, basal cells appear relatively homogeneous, suggesting that previously reported basal heterogeneity (*Goldstein et al., 2008*; *Lawson et al., 2007*; *Wang et al., 2020*) may be more limited. Notably, the PrU population is not readily classified in either compartment as it is comprised of cells with both basal and luminal features.

Our findings also provide a broader context for other reports of epithelial heterogeneity. Recent single-cell RNA-seq analysis of the mouse anterior prostate identified three distinct luminal populations (*Karthaus et al., 2020*), where L1 appears to correspond to our LumA and L2 to LumP; we also identified a population expressing L3 genes in both mouse and human datasets, but have annotated this as ductus/vas deferens based on marker expression (*Figures 1A* and *4C*), relying on previous findings (*Blomqvist et al., 2006*). Another recent paper based on single-cell analysis of the whole mouse prostate has described three luminal populations, in which Luminal-A appears to correspond to our LumV, Luminal-B to our LumA and LumD, and Luminal-C to our LumP; notably, this analysis used lineage-tracing to demonstrate progenitor properties of distal Luminal-C (LumP) cells (*Guo et al., 2020*). Based on patterns of gene expression, we suggest that the LumP population corresponds to the previously reported Sca1$^{high}$ luminal cells (*Kwon et al., 2016*) as well as Trop2 (Tacstd2)-positive cells (*Crowell et al., 2019*), which have been described as proximal progenitor populations with scattered distal cells, and may also be responsible for the enhanced serial grafting efficiency of proximal prostate (*Goto et al., 2006*). In the human prostate, the progenitor activity of LumP and/or PrU-like cells may have been observed by retrospective lineage tracing using mitochondrial mutations (*Moad et al., 2017*). In addition, the LumP and PrU populations may share some similarities with the 'hillock' and 'club' cells originally described in a cellular atlas of the mouse lung (*Montoro et al., 2018*) and reported in the human prostate (*Henry et al., 2018*; *Joseph et al., 2020*) and in benign human prostate organoids (*McCray et al., 2019*), but the precise relationship of these populations is unclear.

Since LumP and PrU cells display multipotent progenitor activity in both organoid formation and tissue reconstitution assays, their spatial distribution may reflect the ability of the prostate to repair itself from a proximal to distal direction in response to extensive tissue damage as well as from distal progenitors in response to more localized injury. Our findings suggest that the novel PrU population in particular may play a role in prostate tissue repair and/or regeneration, consistent with the previous identification of Ly6d-positive cells as a castration-resistant progenitor (*Barros-Silva et al., 2018*). Notably, the ability of *Nkx3-1$^{Cre}$* to lineage-mark both PrU and LumP, but not the urethra, suggests that these cell populations are distinct from the urothelium, despite the molecular similarities between the proximal prostate and the urethra noted in a recent report (*Joseph et al., 2020*). Nonetheless, our results imply that lineage relationships among the tissues derived from the urogenital sinus (*Georgas et al., 2015*) require careful elucidation, since they are of fundamental importance for understanding the genesis of congenital defects in the urogenital system.

Our findings help resolve a long-standing question about the relationship of the mouse and human prostates. Specifically, we speculate that mouse proximal and distal regions are most related to human ductal and acinar regions, respectively, and that the mouse LP is most similar to the human peripheral zone. Few if any studies have specifically assessed tumor phenotypes in the LP, as it is small and usually combined with the DP as the dorsolateral prostate (DLP); however, our analyses show that the DP differs significantly from the LP at the anatomical and molecular levels. Consequently, we suggest that a re-evaluation of tumor phenotypes in genetically engineered mouse models may reveal a closer similarity to human prostate tumor histopathology than previously appreciated.

Finally, the elucidation of prostate epithelial heterogeneity has potentially significant implications for understanding the cell of origin for prostate adenocarcinoma. Previous studies have suggested that luminal cells as well as basal cells can serve as the cell of origin for prostate cancer (*Wang et al., 2013*; *Wang et al., 2014*; *Xin, 2019*), yet known differences in human prostate cancer outcome (e.g. *Zhao et al., 2017*) cannot be simply explained on this basis. Notably, previous studies suggest that both distal and proximal luminal cells can serve as cells of origin for prostate cancer. For example, prostate tumors arise following deletion of *Pten* by the *Nkx3.1-CreER$^{T2}$* driver, which is specific for distal luminal cells (*Wang et al., 2013*), as well as by the *Ck4-CreER$^{T2}$* driver, which is specific for LumP cells (*Guo et al., 2020*). Therefore, further analyses of epithelial heterogeneity and progenitor potential will likely lead to key insights into prostate tumor initiation and progression.

# Materials and methods

## Key resources table

| Reagent type (species) or resource | Designation | Source or reference | Identifiers | Additional information |
|---|---|---|---|---|
| Strain, strain background (*Mus musculus*) | C57BL/6N (wild type) | Taconic | C57BL/6NTac | 8- to 10-week-old males |
| Strain, strain background (*Mus musculus*) | SW (wild type) | Taconic | Tac:SW | 8- to 10-week-old males |
| Strain, strain background (*Mus musculus*) | *UBC-GFP* | Jackson Laboratory, JAX #004353 | C57BL/6-Tg(UBC-GFP)30Scha/J | BL6 background, 8- to 13-week-old males |
| Strain, strain background (*Mus musculus*) | *R26R-YFP* | Jackson Laboratory, JAX #007903 | B6.Cg–*Gt(ROSA)26Sor$^{tm3(CAG-EYFP)Hze}$*/J | 8- to 13-week-old males |
| Strain, strain background (*Mus musculus*) | *Nkx3-1$^{Cre}$* | Shen lab | | BL6 background, 8- to 13-week-old males |
| Strain, strain background (*Mus musculus*) | R2G2 | Envigo | B6;129- *Rag2$^{tm1Fwa}$ Il2rg$^{tm1Rsky}$*/DwlHsd | 8- to 15-week-old males |
| Strain, strain background (*Mus musculus*) | NOD/SCID | Jackson Laboratory, JAX #001303 | NOD.Cg-*Prkdc$^{scid}$*/J | 8- to 15-week-old males |
| Strain, strain background (*Rattus norvegicus domestica*) | Sprague-Dawley embryos | Charles River #400 | SAS Sprague Dawley | E18 embryos from pregnant females |
| Antibody | Anti-mouse Cd66a (CEACAM1)-PE (recombinant monoclonal) | Miltenyi | cat 130-106-209, lot 5190208411, clone REA410 | FACS (1:700) |
| Antibody | Anti-mouse Trop-2 (Tacstd2)-APC (goat polyclonal) | R and D | cat FAB1122A, lot AAZB0117091 | FACS (1:200) |
| Antibody | Anti-mouse Sca-1 (Ly-6a/e) PE-Vio770 (recombinant monoclonal) | Miltenyi | cat 130-106-258, lots 5190308494 and 5180615495, clone REA422 | FACS (1:700) |
| Antibody | Anti-mouse Cd31 (Lin)-FITC (rat monoclonal) | eBiosciences | cat 11-0311-82, lot 1978184, clone 390 | FACS (1:700) |

*Continued on next page*

*Continued*

| Reagent type (species) or resource | Designation | Source or reference | Identifiers | Additional information |
|---|---|---|---|---|
| Antibody | Anti-mouse Cd45 (Lin)-FITC (rat monoclonal) | eBiosciences | cat 11-0451-82, lot 2015744, clone 30-F11 | FACS (1:700) |
| Antibody | Anti-mouse Ter119 (Lin)-FITC (rat monoclonal) | eBiosciences | cat 11-5921-82, lot 2009756, clone TER-119 | FACS (1:700) |
| Antibody | Anti-mouse Ly6d -APC-Vio770 (recombinant m onoclonal) | Miltenyi | cat 130-115-315, lot 5190715088, clone REA A906 | FACS (1:700); IF (1:100) |
| Antibody | Anti-mouse Ppp1r1b (DARPP-32) (mouse monoclonal) | SCBT | cat sc-271111, lot C2719, clone H-3 | IF (1:50) |
| Antibody | Anti-mouse Ppp1r1b (DARPP-32) (rabbit monoclonal) | Invitrogen | cat MA5-14968, lot VB2947074 | IF (1:400) |
| Antibody | Anti-mouse Trpv6 (rabbit polyclonal) | Alomone labs | cat ACC-036, lot ACC036AN1002 | IF (1:100) |
| Antibody | Anti-mouse Lrrc26 (rabbit polyclonal) | Alomone labs | cat APC-070, lot APC070AN0102 | IF (1:100) |
| Antibody | Anti-mouse Msmb (rabbit polyclonal) | Abclonal | cat A10092, lot 204440101 | IF (1:100) |
| Antibody | Anti-mouse Cldn10 (rabbit polyclonal) | Invitrogen | cat 38–8400, lot UA279882 | IF (1:100) |
| Antibody | Anti-mouse Mgll (rabbit polyclonal) | Invitrogen | cat PA5-27915, lots TI2636340A and VB2935243A | IF (1:250) |
| Antibody | Anti-mouse Tgm4 (rabbit polyclonal) | Invitrogen | cat PA5-42106, lot uc2737144 | IF (1:100) |
| Antibody | Anti-mouse Gsdma (rabbit monoclonal) | Abcam | cat ab230768, lot GR3212791-1 | IF (1:100) |
| Antibody | Anti-mouse Krt7 (rabbit monoclonal) | Abcam | cat ab68459, lot GR40294-6 | IF (1:250–500 uL) |
| Antibody | Anti-mouse Aqp3 (rabbit polyclonal) | Biorbyt | cat orb47955, lot B3440 | IF (1:500) |
| Antibody | Anti-mouse Krt5 (chicken polyclonal) | Biolegend | cat 905901, lot B271562 | IF (1:500) |
| Antibody | Anti-mouse p63 (rabbit polyclonal) | Biolegend | cat 619002, lot B262186 | IF (1:250) |
| Antibody | Anti-mouse Krt8/18 (rat monoclonal) | DSHB | Troma-1 | (1:250, lot-specific) |
| Antibody | Anti-mouse Synaptophysin (mouse monoclonal) | BD Biosciences | cat BD611880, lot 8290534 2 | IF (1:500) |
| Antibody | Anti-mouse Chromogranin A (rabbit polyclonal) | Abcam | cat ab15160, lot GR3205971-2 | IF (1:500) |
| Antibody | Anti-GFP (chicken polyclonal) | Abcam | cat ab13970, lot GR3190550-30 | IF (1:1000) |
| Antibody | Anti-mouse Nkx3.1 (rabbit polyclonal) | Athena Enzymes | cat 0315, lot 20316 | IF (1:100) |
| Antibody | Anti-mouse Ki67 (rabbit polyclonal) | Abcam | cat ab15580, lot GR3198158 | IF (1:100) |
| Antibody | Anti-mouse Krt4 (mouse monoclonal) | Invitrogen | cat MA1-35558, lot TB2524522 | IF (1:100) |

*Continued on next page*

*Continued*

| Reagent type (species) or resource | Designation | Source or reference | Identifiers | Additional information |
|---|---|---|---|---|
| Antibody | Anti-mouse Clusterin (rabbit polyclonal) | LS-Bio | cat LS-331486, lot 115142 | IF (1:100) |
| Antibody | Anti-mouse Wfdc2 (rabbit polyclonal) | Invitrogen | cat PA5-80226, lot TK2671201 | IF (1:100) |
| Antibody | Anti-mouse Ar (Androgen receptor) (rabbit monoclonal) | Abcam | cat ab133273, lot GR3271456-1 | IF (1:100) |
| Antibody | Anti-human Krt7 (mouse monoclonal) | Thermo Fisher | cat MA1-06316, lot OVTL12/30 | IF (1:200) |
| Antibody | Anti-human Rarres1 (mouse monoclonal) | Thermo Fisher | cat MA5-26247, lot OTI1D2 | IF (1:200) |
| Antibody | Anti-human *Cd10* (Mme) (mouse monoclonal) | SCBT | cat sc-46656, clone F-4 | IF (1:100) |
| Antibody | Anti-human Msmb (rabbit polyclonal) | Abclonal | cat A10092 | IF (1:200) |
| Software, algorithm | Random Matrix Theory | R. Rabadan Lab | | https://rabadan.c2b2.columbia.edu/html/randomly/ |
| Software, algorithm | Python Optimal Transport | Rémi Flamary and Nicolas Courty, POT Python Optimal Transport library | | https://github.com/rflamary/POT |
| Software, algorithm | Phylogenetic tree analysis | Phangorn package | | https://github.com/KlausVigo/phangorn |
| Software, algorithm | Leiden algorithm | F. A. Wolf, P. Angerer, and F. J. Theis, Genome Biology (2018). 'SCANPY: large-scale single-cell gene expression data analysis' | | https://scanpy.readthedocs.io/en/stable |

## Mouse strains and genotyping

Wild type C57BL6/N (C57BL/6NTac, 8–10 weeks old) and Swiss-Webster (8–10 weeks) mice were purchased from Taconic. The *UBC-GFP* (C57BL/6-Tg(UBC-GFP)30Scha/J, 8–13 weeks old; JAX #004353)(*Schaefer et al., 2001*) and *R26R–YFP* (B6.Cg–Gt(ROSA)26Sor$^{tm3(CAG-EYFP)Hze}$/J; JAX #007903) (*Madisen et al., 2010*) mice were obtained from the Jackson Laboratory. The *Nkx3-1$^{Cre}$* allele has been previously described (*Lin et al., 2007*; *Thomsen et al., 2008*). As hosts for renal grafts, R2G2 mice (B6;129- *Rag2$^{tm1Fwa}$Il2rg$^{tm1Rsky}$*/DwlHsd, 8–15 weeks old) were purchased from Envigo, and NOD/SCID mice (NOD.Cg-*Prkdc$^{scid}$*/J, 8–14 weeks old; JAX #001303) were purchased from the Jackson Laboratory. To obtain urogenital mesenchyme for tissue recombination and renal grafting, we used E18 Sprague-Dawley rat embryos from timed matings (Charles River #400). Animal studies were approved by and conducted according to standards set by the Columbia University Irving Medical Center (CUIMC) Institutional Animal Care and Use Committee (IACUC).

## Isolation of mouse prostate tissue

The anterior (AP), dorsal (DP), lateral (LP), and ventral prostate (VP) lobes were dissected individually, using a transverse cut at the intersection of each lobe with the urethra to include the periurethral (PrU) region. For some analyses, we dissected PrU tissues in the most proximal regions (extending 0.5–2 mm from the connection of the lobes with the urethra), or the remainder of the proximal regions separately from the distal lobes. Paired lobes were collected from a single C57BL/6 mouse for each scRNA-seq experiment, or from 2 to 5 C57BL/6 mice for flow sorting, organoid culture, and tissue reconstitution experiments. For analyses of prostate anatomy, lobes were either dissected individually or a deep cut was made at the caudal end of the urethra for removal of entire urogenital apparatus.

## Acquisition and pathological assessment of human prostate tissue samples

Human prostate tissue specimens were obtained from patients undergoing cystoprostatectomy for bladder cancer or radical prostatectomy at Columbia University Irving Medical Center or at Weill Cornell Medicine. Patients were aged 54–79 years old and gave informed consent under Institutional Review Board-approved protocols. The clinical characteristics of these patients are provided in *Figure 4—source data 1*. Following surgery, prostate tissue was submitted for gross pathological annotation and sectioning, with ischemic time less than 1 hr.

To acquire samples for single-cell RNA-sequencing, the prostate was transversely sectioned perpendicular to the urethra in three main parts (apex, mid and base), which were further divided based on laterality (left or right). Each part was cut in thick sections that included all three prostatic zones (peripheral, transversal and central). Thick sections with low or no tumor burden were selected for the study based on clinical findings and/or biopsies, and divided in three plates by performing two parallel cuts. The upper flanking plate was flash-frozen, cryosectioned, and a rapid review was performed by a board-certified surgical pathologist (H.H.) to provide preliminary assessment on the presence of benign prostate tissue/absence of carcinoma. The middle flanking plate was stored in RPMI medium with 5% FBS on ice, and immediately transferred to the research facility for single-cell RNA sequencing. The lower flanking plate was processed by formalin fixation and paraffin embedding, followed by sectioning and histological review to confirm presence of benign prostate tissue/absence of carcinoma.

For immunostaining analysis of prostate tissue sections, blocks previously assessed as containing benign prostate histology were selected by a surgical pathologist (B.D.R.). Paraffin sections were immunostained for markers of interest as well as CK5 to confirm the presence of basal cells, and adjacent sections were stained by H and E. H and E sections were then reviewed to confirm benign pathology.

## Dissociation of mouse and human prostate tissue

Prostate tissues were minced with scissors and then incubated in papain (20 units/ml) with 0.1 mg/ml DNase I (Worthington LK003150) at 37°C with gentle agitation. After 45 min, samples were gently triturated, then incubated for another 20–45 min in papain as needed. Samples were gently triturated again, followed by quenching of the enzyme using 1 mg/ml ovomucoid/bovine serum albumin solution with 0.1 mg/ml DNase I (Worthington LK003150). Cells were passed through a 70 μm strainer (pluriSelect 43-10070-70) and washed with PBS-EDTA with 0.1 mg/ml DNase I (Corning MT-46034CI). If needed, the samples were additionally digested in TrypLE Express (Invitrogen 12605–036) for 3–5 min at 37°C with gentle agitation. The samples were gently triturated and the TrypLE was inactivated by addition of HBSS with 10% FBS and 0.1 mg/ml DNase I. Samples were passed through a 20 μm strainer (pluriSelect 43-10020-70), washed in 1x PBS, and resuspended in appropriate buffers for downstream analyses.

## Single-cell RNA-sequencing

Dissociated cells were washed twice in 1x PBS, passed twice through 20 μm strainers (pluriSelect 43-10020-70), and counted using a hemocytometer or the Countess II FL Automated Cell Counter (ThermoFisher). If the viability of samples was >80% and the single-cell fraction was >95%, the cells were resuspended in 1x PBS with 0.04% BSA at approximately $1 \times 10^6$ cells/ml. Samples were submitted to the Columbia JP Sulzberger Genome Center for single-cell RNA-sequencing on the 10x Genomics Chromium platform. Libraries were generated using the Chromium Single Cell 3' Reagent Kit v2, with 12 cycles for cDNA amplification and 12 cycles for library construction. Samples were sequenced on a NovaSeq 6000 (r1 = 26, i1 = 8, r2 = 91). Sequencing data were aligned and quantified using the Cell Ranger Single-Cell Software Suite (v.2.1.1) with either the GRCm38 mouse or the GRCh38 human reference genomes.

For the mouse prostate, two independent biological replicate samples of whole prostate were submitted for scRNA-seq, with 2361 and 2,927 cells sequenced. Two separate biological replicates for individual lobes were also used for scRNA-seq, with 1,581–2,735 cells sequenced for sample. For human prostate (*Figure 4—source data 1*), three independent samples (#1–3) that corresponded to regions of benign histology were submitted for scRNA-seq, with 1,600, 2,303, and 2,825 cells

sequenced, respectively. Single-cell datasets have been deposited in GEO under accession number GSE150692, and can also be accessed through the Broad Institute Single-Cell Portal (www.singlecell. broadinstitute.org).

## Flow cytometry

Dissociated cell suspensions were counted and resuspended in FACS buffer (1–3% FBS in 1x PBS or HBSS, 1 mM EDTA, and 0.1 mg/ml DNase I). Primary antibodies were added at the final concentrations indicated in the Key Resources Table and incubated at 4°C for 20–30 min. Samples were additionally incubated with 1 µM propidium iodide for 15 min before sorting for dead cell exclusion. Cells were sorted using a BD Influx or Aria, using the widest nozzles and lowest pressure settings (140 µm nozzle and 7 psi for the Influx; 130 µm nozzle and 8 psi for the Aria). Data analysis was conducted using FCS Express seven software. A slightly modified strategy was used to sort prostate cells from *UBC-GFP* mice.

## Organoid culture

For ENR conditions (*Karthaus et al., 2014*), sorted cells from *UBC-GFP* mice were resuspended at a final concentration of 1,000 cells per 30 µl droplet of Matrigel (Corning 354234), and placed in individual wells of a 24-well plate. The Matrigel was covered with 500 µl of ENR medium, supplemented with 10 nM dihydrotestosterone (DHT), 5 mM A 83-01, and 10 µM Y-27632. Media were replenished at day five and organoids were imaged at day 10. Organoid measurements were performed using the Fiji Particle Analysis Plugin (*Rueden et al., 2017*; *Schindelin et al., 2012*), excluding particles with area <2000 $\mu m^2$ and roundness value <0.5. If needed, Watershed was applied to separate overlapping organoids/particles, with wells from at least four independent experiments analyzed.

For hepatocyte media (HM) conditions (*Chua et al., 2014*), sorted cells from wild-type C57BL/6 mice were plated at 2000 and 5,000 cells per well in ultra-low attachment 96-well plates (Corning 3474), and grown in hepatocyte media with 5% Matrigel, 10 nM DHT, and 10 µM Y-27632, with media replenished every 5 days. Organoid formation efficiencies were calculated on day 12–13 of culture. Since LumV cells tended to form small structures containing 1–4 cells, we used a required minimum cut-off size; organoid images were analyzed using ImageJ. Data were collected from three biological replicate experiments, with a minimum of 2–3 technical replicates for each population in each experiment.

## Renal grafting

For tissue reconstitution experiments, urogenital mesenchyme (UGM) cells were collected from embryonic day 18.5 rat embryos as described (*Chua et al., 2018*) and passed through a 100 µM filter (Pluriselect) before use. Sorted mouse epithelial cells were used at ranges from 250 to 60,000 cells depending on the specific population; since basal cells have been previously examined in graft experiments (e.g., *Wang et al., 2013*), we did not investigate the minimum number of basal cells required for graft growth in these experiments. Rat UGM cells and sorted mouse epithelial cells were combined at pre-determined ratios (e.g., 250,000 UGM/5,000 LumP cells) and resuspended in 10–15 µl buttons composed of 9:1 collagen 1 (Corning 354249):setting solution (10x EBSS, 0.2 M NaHCO$_3$, and 50 mM NaOH). After solidification of the collagen, the buttons were incubated in DMEM media with 10% FBS and 10 nM DHT overnight, followed by grafting under the kidney capsule of host immunodeficient mice on the next day. At the time of surgery, a slow-release testosterone pellet (12.5 mg testosterone, 90 day release; Innovative Research of America NA-151) was inserted subcutaneously in each host mouse. Grafts were analyzed 8–12 weeks after surgery.

## Histology, immunostaining, and image analysis

For generation of paraffin blocks, prostate tissues were dissected in ice-cold HBSS and fixed in 10% formalin overnight, followed by processing through an ethanol gradient and embedding. For generation of frozen blocks, dissected tissues were fixed in 4% paraformaldehyde, immersed in sucrose overnight (30% in 1x PBS), embedded in OCT (Tissue-Tek 4583), and stored at −80°C. Alternatively, samples were flash-frozen in 2-methyl-butane (Sigma-Aldrich M32631) at −150°C for 1 hr, then stored at −80°C. Paraffin-embedded tissues were sectioned using a MICROM HM 325 microtome,

and cryo-preserved tissues were sectioned using a Leica CM 1900 cryostat, at thicknesses of 5–13 µm.

For histological analyses, hematoxylin-eosin (H and E) staining was performed on paraffin sections using standard procedures. For immunofluorescence staining of paraffin sections, antigen retrieval was performed by boiling slides in citrate-based or Tris-based antigen unmasking buffer (Vector Labs H3300 and H3301) for 15-45 min. For immunofluorescence of cryosections, slides were rapidly fixed in either 4% paraformaldehyde or 10% NBF for 5 min after sectioning. Slides were washed, blocked in 5% animal serum for 1 hr, and incubated with primary antibodies overnight at 4°C. Slides were washed and incubated with Alexa Fluor secondary antibodies (Life Technologies) for one hour. Sections were stained with DAPI, and mounted (Vector Labs H-1200). Fluorescent images were acquired using a Leica TCS SP2, a Leica TCS SP5, or a Nikon Ti Eclipse inverted confocal microscope.

## Electron microscopy

Prostate tissue from a C57BL/6 mouse at 8 weeks of age was dissected and fixed for 2 hr in 0.1 M sodium cacodylate buffer (pH 7.2) containing 2.5% glutaraldehyde and 2% paraformaldehyde. A portion of the AP lobe at the proximal-distal boundary was micro-dissected and post-fixed for two hours with 1% osmium tetroxide, contrasted with 1% aqueous uranyl acetate, dehydrated using an ethanol gradient and embedded in EMbed 812 (Electron Microscopy Sciences, Hatfield, PA). 70 nm ultrathin sections were cut, mounted on formvar coated slot copper grids, and stained with uranyl acetate and lead citrate. Stained grids were imaged with a Zeiss Gemini300 scanning electron microscope using the STEM detector.

## Statistical analysis

Prism v.8 was used for statistical analyses of functional data and for plot generation. For analyses of organoid formation efficiencies, the data passed the Shapiro-Wilk test for normal distribution ($p >$ alpha = 0.05), but did not pass the Bartlett's test for equal variance ($p < 0.05$). We therefore used the Brown-Forsythe and Welch One-way Analysis of Variance (ANOVA) to confirm statistically significant differences between organoid populations ($p < 0.0001$ for both HM and ENR conditions). Since all populations have fewer than 50 data points per sample, we used the Dunnett's T3 multiple comparisons test to determine which populations significantly differ ($p < 0.05$). The p-values on the graphs indicate the least significant difference observed between compared populations.

For analyses of graft efficiency, the data did not pass the Shapiro-Wilk test for normal distribution ($p < a$ = 0.05), or Bartlett's test for equal variance ($p < 0.05$). We therefore used a non-parametric Kruskal-Wallis test to confirm statistically significant differences between graft input cell numbers by epithelial population ($p < 0.0001$), followed by the two-stage linear step-up procedure of Benjamini, Krieger and Yekutieli ($p < 0.05$). The p-value on the graph indicates the average significant difference between the up to 10 lowest input cell numbers for each distal luminal population compared to the LumP population.

## Bioinformatic analysis of single-cell RNA-seq data

### Outline of analytical pipeline

We first present an overview of our analytical workflow for processing the single-cell RNA-seq data, and describe these steps in detail below. All of the code used in our analysis is publicly available, and links to these packages are provided.

1. Filtering of raw sequencing data
2. Batch effect correction (for aggregation of datasets; https://satijalab.org/seurat/) and processing of data using *Randomly* (https://rabadan.c2b2.columbia.edu/html/randomly/)
3. Clustering of data using the Leiden algorithm (https://scanpy.readthedocs.io/en/stable)
4. Dimensional reduction for visualization by tSNE and UMAP plots (incorporated as part of the *Randomly* package)

### Filtering the expression matrix

The starting pool of cells in the mouse prostate analysis is 13,429 cells, which is composed of two whole prostate samples of 2361 and 2927 cells, and 4 samples corresponding to each of the lobes

at 2735 (AP), 1781 (DP), 2044 (LP), and 1581 (VP) cells. The starting pool of cells for human prostate analyses is 6728 cells coming from three independent samples of 2303, 1600 and 2825 cells each. When filtering the data, we removed cells with less than 500 genes detected and cells with >10% of total transcripts derived from mitochondrial-encoded genes. The expression matrices are normalized by $log_2(1 + TPM)$, where $TPM$ denotes transcripts per million.

## Batch effect correction

For *Figure 1A*, we have aggregated the two samples corresponding to the whole mouse prostate. As a first step to remove batch effects, we used the algorithm described in *Stuart et al., 2019*, using default parameters.

## Random matrix theory application to single-cell transcriptomics

To process our single-cell data, we have used *Randomly,* an algorithm based on Random Matrix Theory (RMT) (*Aparicio et al., 2020*). This algorithm is a public Python package and can be found in https://rabadan.c2b2.columbia.edu/html/randomly/. The idea of this algorithm is based on the fact that a single-cell dataset shows a threefold structure: a random matrix, a sparsity-induced (fake) signal and a biological signal. Indeed, 95% or more of the single-cell expression matrix is compatible with being a random matrix and hence, in such a case, with being pure noise (*Aparicio et al., 2020*). In order to detect the part of the expression matrix compatible with noise, *Randomly* uses the universality properties of RMT. More specifically, let us suppose a $N \times P$ expression matrix $X$, where N is the number of cells and P is the number of genes, and where each column is independently drawn from a distribution with mean zero and variance $\sigma$, the corresponding Wishart matrix is defined as an $N \times N$ matrix:

$$W = \frac{1}{P} XX^T$$

The eigenvalues $\lambda_i$ and normalized eigenvectors $\psi_i$ of the Wishart matrix where $i = 1, 2, \ldots N$ are given by the following relation:

$$W\psi_i = \lambda_i \psi_i$$

If *X* happens to be a random matrix (a matrix whose entries $x_{ij}$ are randomly sampled from a given distribution), then *W* becomes a random covariance matrix and the properties of its eigenvalues and eigenvectors are described by Random Matrix Theory. Universality properties of RMT arise in the limit $N \to \infty, \; P \to \infty, \; \gamma = \frac{N}{P}$ fixed. One of the consequences of universality at the level of eigenvalues $\lambda_i$, is that empirical density of states converges to the so-called Marchenko-Pastur (MP) distribution:

$$\rho_{MP}(\lambda) = \frac{1}{2\pi\gamma\sigma^2} \frac{\sqrt{(a_+ - \lambda)(\lambda - a_-)}}{\lambda} I_{[a_-, a_+]}$$

where

$$a_\pm = \sigma^2 \left(1 \pm \sqrt{\gamma}\right)^2$$

and $\sigma$ represents the variance of the probability distribution that generates each element in the random matrix ensemble. Any deviation of the eigenvalues from MP distribution would imply that the expression matrix *X* is not completely random, and therefore contains a signal that could be further analyzed.

One of the key features of the *Randomly* algorithm is the analysis of eigenvectors. At the level of eigenvectors, RMT universality is manifested through the so-called eigenvector delocalization, which implies that the norm of the eigenvectors $\psi_i$ is equally distributed among all their components α:

$$\left|\psi_i^{(\alpha)}\right| \sim \frac{1}{\sqrt{N}}$$

Interestingly, the distribution of components for delocalized eigenvectors at large N approximates a Gaussian distribution with mean zero and 1/N variance

$$f(\psi) \sim \frac{N}{\sqrt{2\pi}} e^{\left(\frac{-N\psi^2}{2}\right)}$$

The presence of any localized (non-delocalized) eigenvector implies that expression matrix $X$ is not completely random, and hence the existence of a signal that carries information.

However, in single-cell datasets, there is a very important subtlety due to the sparsity, which can generate a fake signal (*Aparicio et al., 2020*). At the single-cell analysis level, the presence of localized eigenvectors related with sparsity implies the existence of an undesired (fake) signal. The *Randomly* algorithm is able to eliminate the sparsity-induced (fake) signal and isolate the biological signal.

In *Figure 1—figure supplement 2*, we show one example of the performance of the *Randomly* algorithm. *Figure 1—figure supplement 2E* shows a comparison of the eigenvector localization with and without sparsity-induced signal in one of the single-cell mouse datasets. The number of eigenvectors that carries signal is larger for the case with sparsity-induced signal. After removal of the fake signal due to sparsity, *Figure 1—figure supplement 2F* shows the distribution of eigenvalues and the fraction behaving in agreement with the MP distribution. More than 95% of the expression matrix is compatible with a random matrix and therefore is equivalent to random noise. In *Figure 1—figure supplement 2G and H*, the algorithm projects the original expression matrix into the signal-like eigenvectors and the noise-like eigenvectors and performs a chi-squared test for the variance (normalized sample variance), which allows identification of the signal-like genes based on a false discovery rate.

## Clustering

Clustering was performed using the Leiden algorithm (*Wolf et al., 2018*), with the number of clusters based on the mean silhouette score. More specifically, we performed a set of clustering performances for different Leiden resolution parameters and computed the mean silhouette score for each case. The silhouette coefficient for a specific cell is given by:

$$s = \frac{b - a}{\max(a, b)}$$

where the $a$ is the mean distance between a cell and all the other cells of the same class, and parameter $b$ is the mean distance between a cell and all other cells in the next nearest cluster. *Figure 1—figure supplement 2I* shows the mean silhouette score as a function of the Leiden resolution parameter and the number of clusters for each case. The strategy we followed was to select the number of clusters that maximizes this correlation. In some cases, it could be also useful to sub-cluster some of the clusters, repeating the strategy just described for one specific cluster. The sub-clustering was used to disentangle the immune populations or the vas deferens and the seminal vesicle populations.

## t-SNE and UMAP representations

In order to visualize single-cell clusters, we performed a further dimensional reduction to two dimensions by t-SNE (t-distributed Stochastic Neighbor Embedding) or UMAP (Uniform Manifold Approximation and Projection) representations, using default parameters. The tSNE, UMAP, dot-plots, and ridge-plots were generated using the visualization functions of the *Randomly* package (*Aparicio et al., 2020*).

## Comparison of RMT with traditional pipelines based on PCA dimensional reduction

To show a comparison with traditional approaches based on PCA, we have followed the pipeline in a public tool (*Wolf et al., 2018*) often used for single-cell analysis. We have performed a PCA reduction, selecting principal components (PCs) through accumulated variance changes across the different PCs (*Figure 1—figure supplement 2A*). In this case, only 10 PCs are selected following this approach. After the dimensional reduction, we performed a clustering following the strategy described in the previous section (*Figure 1—figure supplement 2B*), selecting the number of clusters that maximize the mean silhouette score. Comparing *Figure 1—figure supplement 2B and I*, it

is clear that the RMT generates a better clustering performance: the maximum of the silhouette score curve is larger than that generated by the traditional PCA approach, and one of these clusters is able to capture the periurethral (PrU) population (*Figure 1—figure supplement 2J*). On the other hand, the method based on traditional PCA is not able to capture the PrU population even if we allow for larger Leiden resolutions (*Figure 1—figure supplement 2C and D*).

## Differential expression analysis

To test for differentially expressed genes among the different populations of prostate luminal cells, we used a t-test on the datasets after de-noising them with *Randomly*. The p-value was corrected for multiple hypothesis testing using the Benjamini-Hochberg procedure. We used the implementation of *Wolf et al., 2018* with overestimation of the variance and comparison with a Wilcoxon test. Based on this analysis, we selected genes with a corrected p-value smaller than 0.001.

## Mouse population similarity

To calculate the phenotypic similitude/distance between epithelial populations in the mouse prostate, we performed an analysis based on Optimal Transport (OT) (*Kolouri et al., 2017*; *Villani, 2003*). More specifically, we used the Wasserstein-1 distance as a measure for phenotypic distance between cell populations, that is, among clusters in the latent space obtained after using *Randomly*. The Wasserstein-1 distance is defined as a distance function between probability distributions in a given metric space. Assuming that the metric space is Euclidean, the Wasserstein-1 distance (*Kolouri et al., 2017*; *Villani, 2003*) is defined as:

$$W_1(\mu, \nu) = min\left\{ \int_{\mathbb{R}^d \times \mathbb{R}^d} \|y - x\| \ \gamma(dx, dy): \gamma \in couplings(\mu, \nu) \right\}$$

It is also known as the earth mover's distance, in which each probability distribution can be seen as an amount of dirt piled in the metric space, with the Wasserstein distance corresponding to the cost of turning one pile into the other. In our case, we would be evaluating the cost of transforming one population into another.

The optimization of the Wasserstein calculation can be turned into an OT problem, based on the Sinkhorn algorithm and the entropic regularization technique (*Altschuler et al., 2017*; *Chizat et al., 2018*; *Cuturi, 2013*; *Schmitzer, 2016*). We have used the Python implementation of the package POT Python Optimal Transport library (https://github.com/rflamary/POT), which solves the entropic regularization OT problem and return the loss ($W_1(\mu, \nu)$). We have used as metric cost matrix a Euclidean pairwise distance matrix and assumed that the cell populations correspond to uniform probability distributions defined in the latent space obtained after using *Randomly*.

To calculate the distances between populations, we constructed a matrix of Wasserstein distances among the epithelial populations described in *Figure 1* and visualized it using a heatmap and a hierarchical clustering (*Figure 3A*). We also generated a tree-like visualization of all the information contained in the hierarchical clustering/heatmap using a neighbor joining algorithm (*Schliep, 2011*). The length of the branches in the tree is measured in units of the Wasserstein-1 distance (*Figure 3B*).

## Cross-species analysis

We performed a comparison between the epithelial populations in human and mouse based on OT and Wasserstein distance. To harmonize the human and mouse datasets, we first constructed a common latent space between the aggregated mouse data set and each of the three human samples. To that end, we first looked for the mouse orthologous genes, and then normalized mouse and human separately using $log_2(1 + TPM)$. We filtered out any gene which has an average expression smaller than 0.1 for human or mouse, and merged the two corresponding human and mouse datasets. Finally, we used *Randomly* to generate the common latent space.

We used the Wasserstein distance to calculate the similitude among the clusters of points previously identified with the different mouse and human populations in *Figures 1* and *4*. We visualized this with a set of nested heatmaps (*Figure 4D–F*) to make explicit which populations have the minimum Wasserstein distance between each human population and mouse populations.

We then validated the accuracy of this strategy. The first validation test is that the conserved epithelial populations Basal and Lum P in human have a minimum in the Wasserstein distance with their

mouse equivalents. A second test of the robustness is to compare cell types that are known to be well-conserved across species, such as immune cells. As with the conserved epithelial cell types, the human immune cell populations also have a minimum Wasserstein distance with respect to the corresponding mouse immune populations.

## Acknowledgements

We thank Reuben Akabas, Sarah Bergren, Mykola Bordyuh, Eva Leung, Bo Li, Maximilian Marhold, Agnieszka Pastula, Luis Pina, Roxanne Toivanen, Zixuan Wang, and Sven Wenske for their assistance with this project, and Cory Abate-Shen and Jia Li for comments on the manuscript. We thank Erin Bush and Peter Sims for assistance with single-cell sequencing in the Columbia Genomics and High Throughput Screening Shared Resource of the Herbert Irving Comprehensive Cancer Center, as well as the resources of the Cancer Center Flow Core Facility, which are supported in part by the Cancer Center Support Grant P30CA013696. We also thank Michael Kissner and Daniel Troast for assistance with flow cytometry in the Columbia Stem Cell Initiative Flow Cytometry Core, which is supported in part by S10OD026845. For assistance with electron microscopy, we thank Alice Liang, Chris Petzold, and Joseph Sall at the New York University Langone Health DART Microscopy Lab, which is partially funded by NYU Cancer Center Support Grant P30CA016087 and by S10OD019974. These studies were supported by grants from the NIH R01CA238005 (MMS), U54CA193313 (RR and MMS), P50CA211024 (ML and MMS), K99CA194287 (MS), by the TJ Martell Foundation (MMS), and by fellowships from the Department of Defense Prostate Cancer Research Program (W81XWH-18-1-0424; FC) and the National Science Foundation (DGE-16-44869; LC).

## Additional information

### Funding

| Funder | Grant reference number | Author |
|---|---|---|
| National Cancer Institute | R01CA238005 | Michael M Shen |
| National Cancer Institute | U54CA193313 | Raul Rabadan<br>Michael M Shen |
| National Cancer Institute | P50CA211024 | Massimo Loda<br>Michael M Shen |
| National Cancer Institute | K99CA194287 | Maho Shibata |
| T.J. Martell Foundation | | Michael M Shen |
| U.S. Department of Defense | Prostate Cancer Research Program W81XWH-18-1-0424 | Francesco Cambuli |
| National Science Foundation | DGE-16-44869 | Laura Crowley |

The funders had no role in study design, data collection and interpretation, or the decision to submit the work for publication.

### Author contributions

Laura Crowley, Conceptualization, Funding acquisition, Investigation, Writing - original draft, Writing - review and editing, Conceptualization (data interpretation and presentation); Investigation (single-cell analysis); Investigation (urogenital anatomy); Investigation (mouse prostate analysis); Investigation (organoid assays); Investigation (renal graft assays); Investigation (statistical analysis); Francesco Cambuli, Conceptualization, Funding acquisition, Investigation, Methodology, Writing - original draft, Writing - review and editing, Conceptualization (data interpretation and presentation); Methodology (flow cytometry); Investigation (single-cell analysis); Investigation (urogenital anatomy); Investigation (mouse prostate analysis); Investigation (flow cytometry); Investigation (organoid assays); Investigation (human prostate analysis); Luis Aparicio, Conceptualization, Investigation, Methodology, Writing - original draft, Writing - review and editing, Conceptualization (data interpretation and presentation); Investigation (single-cell analysis); Investigation (computational analysis);

Maho Shibata, Funding acquisition, Investigation, Writing - review and editing, Investigation (single-cell analysis); Brian D Robinson, Hanina Hibshoosh, Massimo Loda, Resources; Shouhong Xuan, Investigation, Writing - review and editing, Investigation (mouse prostate analysis); Investigation (renal graft assays); Weiping Li, Investigation, Investigation (mouse prostate analysis); Raul Rabadan, Conceptualization, Supervision, Funding acquisition, Writing - review and editing, Conceptualization (study design); Michael M Shen, Conceptualization, Supervision, Funding acquisition, Writing - original draft, Writing - review and editing, Conceptualization (study design); Conceptualization (data interpretation and presentation)

### Author ORCIDs
Laura Crowley ⬤ https://orcid.org/0000-0002-6841-2859
Francesco Cambuli ⬤ http://orcid.org/0000-0002-8237-7121
Shouhong Xuan ⬤ http://orcid.org/0000-0003-0571-7855
Michael M Shen ⬤ https://orcid.org/0000-0002-4042-1657

### Ethics
Human subjects: Human prostate tissue specimens were obtained from patients undergoing cysto-prostatectomy for bladder cancer or radical prostatectomy at Columbia University Irving Medical Center or at Weill Cornell Medicine. Patients gave informed consent under an Institutional Review Board-approved protocol (AAAN8850).

Animal experimentation: Animal studies were conducted according to protocols (AC-AABE0556, AC-AABG0564, AC-AABE5557) approved by the Columbia University Irving Medical Center (CUIMC) Institutional Animal Care and Use Committee (IACUC).

### Decision letter and Author response
Decision letter https://doi.org/10.7554/eLife.59465.sa1
Author response https://doi.org/10.7554/eLife.59465.sa2

## Additional files

### Supplementary files
• Transparent reporting form

### Data availability
Single-cell RNA-sequencing data from this study have been deposited in the Gene Expression Omnibus (GEO) under the accession number GSE150692, and can also be accessed through the Broad Institute Single-Cell Portal (https://singlecell.broadinstitute.org/single_cell).

The following dataset was generated:

| Author(s) | Year | Dataset title | Dataset URL | Database and Identifier |
| --- | --- | --- | --- | --- |
| Shen MM, Aparicio L, Cambuli F, Crowley L, Shibata M | 2020 | Single-cell RNA-seq analysis of mouse and human prostate | https://www.ncbi.nlm.nih.gov/geo/query/acc.cgi?acc=GSE150692 | NCBI Gene Expression Omnibus, GSE150692 |

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
