## [Decision Letter]

**Acceptance summary:**

Your article identifies distinct luminal epithelial and periurethral populations of the adult mouse prostate using a single-cell sequencing approach. Furthermore, you identify the human prostate counterparts to these distinct populations of mouse prostate cells, which together advances the our understanding of the relationship between human and mouse prostate populations.

**Decision letter after peer review:**

Thank you for submitting your article "A single-cell atlas of the mouse and human prostate reveals heterogeneity and conservation of epithelial progenitors" for consideration by *eLife*. Your article has been reviewed by three peer reviewers, including Nima Sharifi as the Reviewing Editor and Reviewer #1, and the evaluation has been overseen by Richard White as the Senior Editor. The following individuals involved in review of your submission have agreed to reveal their identity: Mark Rubin (Reviewer #2); Justin D Lathia (Reviewer #3).

The reviewers have discussed the reviews with one another and the Reviewing Editor has drafted this decision to help you prepare a revised submission.

Summary:

This study is a well-done single-cell comparative atlas of the mouse and human prostate. Their studies were from two whole mouse prostates, 4 samples corresponding to each of the lobes and 3 independent sets of human prostate for comparison. Altogether, they identify lobe-specific luminal epithelial populations (LumA, LumD, LumL, and LumV) in the distal region and the proximally-enriched luminal population (LumP) that is not specific to any lobe and a periurethral population (PrU) that has both basal and luminal features. Organoid studies suggest that LumP and PrU cells are multipotent progenitors. Comparisons to the human prostate suggest that the mouse lateral prostate is most like the human peripheral zone – from which the majority of prostate cancers arise. Altogether, this is generally a well done study. The nature of the study is fundamental, the paper is generally well written and the studies themselves generally well-described. All three reviewers concur about the importance of this body of work.

Essential revisions:

Most of the comments pertain to optimizing the clarity of presentation and data accessibility. Please see the specific reviewer comments below.

1) Title. A title referencing the work as an atlas should make the atlas more publicly accessible and reproducible than an expression matrix. Either remove atlas from the title/text or consider making this group of work more accessible to the community so that it can be utilized as an atlas. One way to do this would be to place these data into a shiny app or some other sort of online single cell viewing portal (UCSC cell browser, or the broad institute single cell portal). This is especially important because you use Randomly and many people will place this data directly into other pipelines such as Seurat and the clustering will be different so people may not be able to query your exact populations without further information from your group. Following this the entirety of the code should be published with this manuscript since a newer method clustering is used and there is no single package that this data is being ran through.

2 Subsection “Distinct luminal epithelial populations in the mouse prostate” – please refer how the samples were dissociated for single cell seq here as this is important for future studies.

3) Subsection “Distinct luminal epithelial populations in the mouse prostate” – The Randomly package is explained and referenced but then in the Materials and methods section there are almost 3 pages about Randomly and the equations used. I believe it is enough to say at which steps you implemented the use of this published package and then move on from here. Additionally, please briefly mention that batch effect correction was done and how. I know this is in the Materials and methods but in general one should be able to read in 2-3 sentences the sequencing depth, method of data alignment, batch correction, use of randomly, and then downstream analysis. Extra detail about each of these steps and the exact settings used if not default settings can be added into the Materials and methods.

4) Subsection “Distinct luminal epithelial populations in the mouse prostate” – – it's my belief that these tSNE plots should be UMAPs to show your clusters better. For example, Figure 1A has intermixing of luminal A and D but viewing this as a tSNE I don't know if these clusters are mixed because they are similar and there is a population of LumD that appear more similar to LumA. There is also a PrU cell that appears in the center of the plot away from the rest of the PrU cells. I think these should be re-graphed as UMAP plots for the whole paper. At the very least they should be in the supplement as UMAPs but I suspect they will represent your data better and you may want to swap them once they are graphed.

5) The description and biology of the cellular constituents of the mouse prostate and comparison to human anatomy are well done. In human, the normal prostate anatomic origin of prostate cancer is well worked out. This group has pioneer several different mouse prostate cancer models. Given the profiling in the current paper, can they comment or show data on the cellular origin of mouse prostate cancer? The genetic models may have divergent origins and may be multifocal; however, one wonders if they bear some resemblance to LumP or PrU cells, at least more so compared to the other cell and anatomic types.

6) Subsection “Spatial localization and morphology of epithelial populations” – "suitable antibodies" is vague and needs to be expanded on what makes them suitable.

7) Subsection “Spatial localization and morphology of epithelial populations” – "specificity" for their markers also should be expanded on, without knockdown and knockout mice you can't prove the antibody is specific. It is simply enough to say that we utilized and stained with these antibodies and demonstrated that there was little or no background on the secondary only controls.

8).Subsection “Functional analysis of epithelial populations” – Figure 3C is unclear what these flow plots actually represent? What was the starting population or flow and what gates preceded these representative panels. Include this in the text and figure legend.

9) Discussion, first paragraph – function properties are noted and it would be nice to see differential gene expression showed for the groups and the significantly expressed differences between groups should be published as supplementary tables.

10) Subsection “Data availability”. Utilizing new methods of single cell analysis and calling the data an atlas and identifying new cell types necessitates that this data be available in a better format than GEO. The entirety of the code along with metadata showing which cells are in which clusters need to be made available. An added bonus would be uploading it to a portal so people can query their own genes with the cell clusters you identified but at the very minimum the above criteria should be met.

11) Referencing/novelty – There are several single cell studies in this general area, but are not referenced, and they should be (PMIDs: 31317052, 30566875, 29233929). It is acknowledged that this study is clear different, but the reader should be made aware of the current state of the field.

---

## [Author Response]

Essential revisions:Most of the comments pertain to optimizing the clarity of presentation and data accessibility. Please see the specific reviewer comments below.1) Title. A title referencing the work as an atlas should make the atlas more publicly accessible and reproducible than an expression matrix. Either remove atlas from the title/text or consider making this group of work more accessible to the community so that it can be utilized as an atlas. One way to do this would be to place these data into a shiny app or some other sort of online single cell viewing portal (UCSC cell browser, or the broad institute single cell portal). This is especially important because you use Randomly and many people will place this data directly into other pipelines such as Seurat and the clustering will be different so people may not be able to query your exact populations without further information from your group. Following this the entirety of the code should be published with this manuscript since a newer method clustering is used and there is no single package that this data is being ran through.

We thank the reviewers for this valuable comment. Since we believe that our study indeed constitutes an “atlas”, we agree that it is appropriate for us to deposit the single-cell RNA-seq data in a suitable on-line portal for ease-of-access. Consequently, we have placed our datasets in the Broad Institute Single Cell Portal, which can be accessed through the following temporary links: https://singlecell.broadinstitute.org/single_cell/study/SCP1080/anterior#study-visualize https://singlecell.broadinstitute.org/single_cell/study/SCP1081/dorsal#study-visualize

https://singlecell.broadinstitute.org/single_cell/study/SCP1082/hum1#study-visualize https://singlecell.broadinstitute.org/single_cell/study/SCP1083/hum2#study-visualize https://singlecell.broadinstitute.org/single_cell/study/SCP1084/hum3#study-visualize https://singlecell.broadinstitute.org/single_cell/study/SCP1085/lateral#study-visualize https://singlecell.broadinstitute.org/single_cell/study/SCP1086/ventral#study-visualize

At present, these datasets can still considered to be in a “beta” format, but we will ensure that they are in a final form and are made available for public access prior to publication. We have noted that the datasets can be accessed at the Broad Institute Single Cell Portal in the subsections “Single-cell RNA-sequencing” and “Data availability”.

With respect to code availability, all of the code used in this study is publicly available and cited appropriately in the Materials and methods. While we did not use a single package to process our data, we have revised the Materials and methods description in the subsection “Outline of analytical pipeline” to outline which packages were used in sequence, and have provided links for access.

2) Subsection “Distinct luminal epithelial populations in the mouse prostate” – please refer how the samples were dissociated for single cell seq here as this is important for future studies.

In the subsection “Distinct luminal epithelial populations in the mouse prostate”, we have added text describing the dissociation protocol, and refer to the Materials and methods for details of the methodology.

3) Subsection “Distinct luminal epithelial populations in the mouse prostate” – The Randomly package is explained and referenced but then in the Materials and methods section there are almost 3 pages about Randomly and the equations used. I believe it is enough to say at which steps you implemented the use of this published package and then move on from here. Additionally, please briefly mention that batch effect correction was done and how. I know this is in the Materials and methods but in general one should be able to read in 2-3 sentences the sequencing depth, method of data alignment, batch correction, use of randomly, and then downstream analysis. Extra detail about each of these steps and the exact settings used if not default settings can be added into the Materials and methods.

In response to this critique, we have added a brief description of the analytical pipeline in the subsection “Distinct luminal epithelial populations in the mouse prostate”, and refer to the Materials and methods for details. As noted in critique #1 above, we have also simplified the description in the Materials and methods to clarify the workflow. However, we have retained most of the text on the *Randomly* algorithm and the accompanying Figure 1—figure supplement 2 since we believe that these descriptions are essential for understanding the advantages of our computational approach.

4) Subsection “Distinct luminal epithelial populations in the mouse prostate” – it's my belief that these tSNE plots should be UMAPs to show your clusters better. For example, Figure 1A has intermixing of luminal A and D but viewing this as a tSNE I don't know if these clusters are mixed because they are similar and there is a population of LumD that appear more similar to LumA. There is also a PrU cell that appears in the center of the plot away from the rest of the PrU cells. I think these should be re-graphed as UMAP plots for the whole paper. At the very least they should be in the supplement as UMAPs but I suspect they will represent your data better and you may want to swap them once they are graphed.

In response to this request, we have generated UMAP plots that correspond to each of the tSNE plots shown in the main figures. These UMAP plots are now shown in new supplementary figures (Figure 1—figure supplement 3 and Figure 4—figure supplement 1). We believe that these UMAP plots are complementary to the tSNE plots shown in the main figures, and that providing both sets of plots may be advantageous to some readers. With respect to Figure 1A, we note that LumA and LumD populations are indeed closely related and slightly intermingled, and remain so in the UMAP representation in the new Figure 1—figure supplement 3A.

5) The description and biology of the cellular constituents of the mouse prostate and comparison to human anatomy are well done. In human, the normal prostate anatomic origin of prostate cancer is well worked out. This group has pioneer several different mouse prostate cancer models. Given the profiling in the current paper, can they comment or show data on the cellular origin of mouse prostate cancer? The genetic models may have divergent origins and may be multifocal; however, one wonders if they bear some resemblance to LumP or PrU cells, at least more so compared to the other cell and anatomic types.

We agree with the reviewers that this is an important topic, and have added additional sentences to the last paragraph of the Discussion to state that both proximal and distal luminal cells can likely serve as cells of origin for prostate cancer. However, a comprehensive analysis of cell types of origin for prostate cancer lies well beyond the scope of the present study.

6) Subsection “Spatial localization and morphology of epithelial populations” – "suitable antibodies" is vague and needs to be expanded on what makes them suitable.

We agree that the previous description was vague, and have now added text to the subsection “Spatial localization and morphology of epithelial populations”, to describe how antibodies were selected for study.

7) Subsection “Spatial localization and morphology of epithelial populations” – "specificity" for their markers also should be expanded on, without knockdown and knockout mice you can't prove the antibody is specific. It is simply enough to say that we utilized and stained with these antibodies and demonstrated that there was little or no background on the secondary only controls.

As noted in our response to critique #6 above, we have now clarified the description of how antibodies were selected in the subsection “Spatial localization and morphology of epithelial populations”.

8) Subsection “Functional analysis of epithelial populations” – Figure 3C is unclear what these flow plots actually represent? What was the starting population or flow and what gates preceded these representative panels. Include this in the text and figure legend.

We agree with the reviewers that our previous description of flow cytometry was inadequate. Consequently, we have added the new Figure 3—figure supplement 1A to show the complete flow sorting strategy, have updated the panels in Figure 3C, and have revised the text in the subsection “Functional analysis of epithelial populations” accordingly. In addition, we now provide a control re-sorting experiment in Figure 3— figure supplement 1B to demonstrate the purity of the cell populations obtained by our sorting strategy.

9) Discussion, first paragraph – function properties are noted and it would be nice to see differential gene expression showed for the groups and the significantly expressed differences between groups should be published as supplementary tables.

In response to this critique, we have included the new Figure 1—figure supplement 4 to show an expanded dot plot for genes that are differentially expressed between the different epithelial populations, based on our single-cell RNA-seq data, and have cited this new figure in the subsection “Spatial localization and morphology of epithelial populations”. We believe that this dot plot is more useful for the reader than additional supplementary tables, since differential expression can be assessed at a glance in Figure 1—figure supplement 4.

10) Subsection “Data availability”. Utilizing new methods of single cell analysis and calling the data an atlas and identifying new cell types necessitates that this data be available in a better format than GEO. The entirety of the code along with metadata showing which cells are in which clusters need to be made available. An added bonus would be uploading it to a portal so people can query their own genes with the cell clusters you identified but at the very minimum the above criteria should be met.

As described in our response to critique #1, we have made all of our datasets available on the Broad Institute single-cell web portal. Importantly, this web portal will allow readers to query our datasets with their genes of interest. Also, as stated above, all of the code utilized in this study is publicly available. We have generated metadata for each dataset, which will be provided in GSE150692.

11) Referencing/novelty – There are several single cell studies in this general area, but are not referenced, and they should be (PMIDs: 31317052, 30566875, 29233929). It is acknowledged that this study is clear different, but the reader should be made aware of the current state of the field.

In response to this critique, we have expanded the Discussion to discuss and cite PMID 31317052 and 30566875, but not PMID 29233929, which is a study of LNCaP prostate cancer cells that we believe is not relevant for our work. We have also cited and described in lines 260-264 the new paper from the Gao laboratory (Guo et al., 2020), which was published recently in Nature Genetics.